# The Association between Pressure Pain Thresholds, Conditioned Pain Modulation, Clinical Status, and Sleep Quality in Fibromyalgia Patients: A Clinical Trial Secondary Analysis

**DOI:** 10.3390/jcm13164834

**Published:** 2024-08-16

**Authors:** María Elena González-Álvarez, Víctor Riquelme-Aguado, Alberto Arribas-Romano, Josué Fernández-Carnero, Jorge Hugo Villafañe

**Affiliations:** 1Escuela Internacional de Doctorado, Rey Juan Carlos University, 28008 Madrid, Spain; 2Department of Physical Therapy, Occupational Therapy, Rehabilitation and Physical Medicine, Rey Juan Carlos University, 28032 Madrid, Spain; alberto.arribas@urjc.es (A.A.-R.); josue.fernandez@urjc.es (J.F.-C.); 3Cognitive Neuroscience, Pain, and Rehabilitation Research Group (NECODOR), Faculty of Health Sciences, Rey Juan Carlos University, 28922 Madrid, Spain; 4Department of Basic Health Sciences, Rey Juan Carlos University, 28933 Madrid, Spain; 5Grupo de Investigación Emergente de Bases Anatómicas, Moleculares y del Desarrollo Humano, Universidad Rey Juan Carlos (GAMDES), 28922 Alcorcón, Spain; 6Fisioterapia Oreka CB, 45200 Illescas, Spain; 7Department of Physiotherapy, Faculty of Sport Sciences, Universidad Europea de Madrid, 28670 Villaviciosa de Odón, Spain; mail@villafane.it; 8Musculoskeletal Pain and Motor Control Research Group, Faculty of Sport Sciences, Universidad Europea de Madrid, 28670 Villaviciosa de Odón, Spain

**Keywords:** fibromyalgia, central sensitization, quantitative sensory test (QST)

## Abstract

**Background**: Fibromyalgia (FM) is a complex multidimensional disorder primarily characterized by chronic widespread pain, significantly affecting patients’ quality of life. FM is associated with some clinical signs found with quantitative sensory testing (QST), sleep disturbance, or psychological problems. This study aims to explore the associations between pressure pain thresholds (PPTs), conditioned pain modulation (CPM), clinical status, and sleep quality in FM patients, offering insights for better clinical management and assessment tools. **Methods**: This secondary analysis utilized data from a clinical trial involving 129 FM patients. Various assessments, including the Fibromyalgia Impact Questionnaire (FIQ), Pain Catastrophizing Scale (PCS), State-Trait Anxiety Inventory (STAI), and Jenkins Sleep Scale (JSS), were employed to evaluate the clinical and psychological status and sleep quality. PPTs and CPM were measured to understand their relationship with clinical parameters. **Results**: Our findings revealed that PPTs and CPM are not significantly associated with the clinical status or sleep quality of FM patients. Instead, pain catastrophizing and anxiety state showed a stronger correlation with the impact of fibromyalgia and sleep disturbances. These results highlight the importance of psychological and cognitive factors in managing FM. **Conclusions**: The study suggests that while PPTs and CPM may not be reliable biomarkers for clinical status in FM, the use of comprehensive assessments including FIQ, PCS, STAI, and JSS can provide a more accurate evaluation of patients’ condition. These tools are cost-effective, can be self-administered, and facilitate a holistic approach to FM management, emphasizing the need for personalized treatment plans.

## 1. Introduction

Fibromyalgia (FM) is a complex multidimensional disorder characterized by pain as the primary symptom. This is accompanied by other equally significant physical, psychological, and cognitive symptoms, including fatigue, non-restorative sleep, psychological symptoms of depression and anxiety, and cognitive distortions such as pain catastrophizing [1,2]. In Madrid, Spain, the prevalence of this syndrome reaches 5% of the population, with women aged between 46 and 60 years being the most affected [3]. Despite extensive research, the aetiology of FM remains elusive, complicating effective clinical management [4]. 

Empirical evidence consistently shows abnormalities in nociceptive transmission and pain modulation pathways in FM patients, indicating an imbalance favouring nociceptive facilitation and resulting in central sensitization [5,6]. Additionally, the COVID-19 pandemic exacerbated emotional and cognitive dysfunctions in FM patients, amplifying pain symptoms due to increased anxiety, depression, healthcare access disruptions, and social isolation [7,8]. These exacerbations highlight the importance of understanding the multifactorial mechanisms underlying FM, which involve genetic, neurobiological, and environmental factors. Genetic studies suggest a heritable component to FM, with familial clustering observed in multiple studies. Polymorphisms in genes related to the serotonergic, dopaminergic, and catecholaminergic systems have been implicated, potentially affecting pain perception and modulation [7]. Environmental triggers such as physical trauma, emotional stress, and certain infections are also frequently reported to precipitate or worsen FM symptoms, underscoring the need for a comprehensive approach to prevention and management [7].

Neuroimaging studies reveal that FM is associated with structural and functional alterations in brain areas responsible for pain processing and modulation, including decreased gray matter volume and altered functional connectivity in pain networks. These neurobiological changes are believed to contribute to the hallmark symptoms of FM, such as chronic widespread pain, fatigue, and emotional disturbances [9].

A key clinical feature of FM is the malfunction of pain inhibitory mechanisms. Quantitative sensory tests (QST) are noninvasive tests of assessing and quantifying the nociceptive system to obtain a more accurate understanding of pain and pain inhibition mechanisms [10]. Conditioned Pain Modulation (CPM) is a physiological assessment measuring the efficacy of descending pain modulatory systems. CPM effectiveness varies significantly among individuals, possibly due to differences in protocol implementation [11,12,13]. Recent evidence suggests that psychological factors do not affect CPM in other pain populations, but CPM research specifically addressing FM patients is limited, leaving a gap in understanding the potential modulatory effects of FM-specific symptoms [14]. 

In FM patients, the impact of clinical status on the outcomes of the CPM test remains uncertain. Numerous variables in FM potentially affecting the outcome include current functional status, sleep quality, and psychological aspects. Previous scientific literature has shown conflicting results regarding correlations between variables. For instance, some studies have found that the presence of depression and anxiety symptoms may influence outcomes, while others have not [15]. The main limitations of these works were the small sample size and the statistical analysis method used. Additionally, other research assessing functional [14] or psychological status has used cutoff points to determine the severity of impairment (mild, moderate, or severe) instead of analyzing the total numerical score obtained. Consequently, there is an imperative need for more robust studies with larger cohorts and enhanced methodological rigour. Notably, the correlation between sleep quality and FM, particularly in the context of its impact on CPM, remains underexplored, with scant literature addressing this critical aspect [16].

Another key clinical characteristic in patients with FM is a reduced pressure pain threshold (PPT) [17] in response to pressure stimuli. This mechanical hyperalgesia is hypothesized to result from peripheral pain transmission alterations. Studies have shown that type C and A-delta fibers exhibit functional and histological changes [18]. These fibers display spontaneous activity, signs of peripheral sensitization, and abnormal latency times after stimulation. It is also suggested that central neuronal hyperexcitability influences the perception of PPT and the amplification of pain signals [19]. However, the impact of these findings on the functional status of fibromyalgia patients remains unknown. Furthermore, previous findings [20] suggest that chronic pain negatively affects sleep quality, creating a vicious cycle where pain interferes with sleep and lack of sleep increases pain perception. However, the relationship between mechanical hyperalgesia and sleep quality in fibromyalgia patients has not been thoroughly studied.

Understanding the mechanisms of pain modulation in FM is crucial for developing targeted therapies. Research into pain modulation pathways, including CPM, has shown that FM patients often exhibit a diminished ability to inhibit pain signals. This impaired pain inhibition, characteristic of central sensitization, suggests that the central nervous systems in FM patients processes pain signals differently from those of healthy individuals [21]. This understanding has paved the way for exploring therapeutic options aimed at enhancing pain inhibition pathways, such as through pharmacological means, neuromodulation techniques, or cognitive–behavioural therapies aimed at altering pain perception [8].

Given the challenges associated with pharmacological treatment of FM, including variable efficacy and potential side effects, non-pharmacological interventions play a pivotal role in managing FM. Exercise, for instance, has been shown to improve pain thresholds and enhance quality of life by reducing stress and improving physical functioning. Cognitive–behavioral therapy (CBT) and other psychological interventions target the emotional and cognitive aspects of pain, helping patients develop coping strategies to manage their symptoms more effectively [22].

The integration of interdisciplinary treatment approaches offers the best promise for the effective management of FM. Tailored treatment plans that consider the individual’s symptoms, lifestyle, psychological state, and physical health are essential. Research is ongoing to better understand the interactions between these various factors and to develop interventions that can modulate pain pathways more effectively.

We hypothesize that fibromyalgia patients with poorer pain modulation capacity, greater mechanical hyperalgesia, and higher levels of anxiety, depression, pain catastrophizing, and fear of movement will correlate with worse functional status and poorer sleep quality. Regarding sleep quality, we hypothesize that it also correlates with the results obtained in CPM, as the lack of restorative sleep can cause a deficit in neurotransmitters involved in pain modulation, such as serotonin. This study aims to determine the relationship between mechanical hyperalgesia, pain modulation, and psychological aspects with functional capacity and sleep quality in patients with FM.

## 2. Materials and Methods

### 2.1. Study Design 

This is a secondary analysis of baseline assessments from two previous randomized controlled trials [23,24]. Both were conducted in accordance with the Helsinki Declaration and approved by the Ethical Committee. The two clinical trials were registered at clinicaltrails.gov (NCT03882567 and NCT02474875). All the participants were informed verbally and in writing prior to the start of the study procedures. 

### 2.2. Participants 

From the cohort of participants generated from both studies, those with no missing data on the main variables of the present study were selected (*n* = 129). Participants were women and were included if they fulfilled the American College of Rheumatology classification criteria for FM [25], reported more than 4 out of 10 on a visual analogy scale, had a stable dose of medication for FM, and were between 18 and 65 years old. The exclusion criteria were: major mental illness (major depression, bipolar disorder, panic disorder, or psychosis), elective surgery during the study period, inflammatory rheumatic condition, and non-Spanish-speaking.

### 2.3. Outcome Measurements

#### 2.3.1. Pressure Pain Threshold (PPT)

PPT is defined as the lowest pressure needed to elicit the first sensation of pain [26]. The PPT was measured using an analogue algometer (Force Dial FDK/FDN 100 model, Wagner Instruments, Greenwich, CT, USA) with a surface of 1 cm^2,^ and the force was applied at a rate of 1 kg/cm^2^. PPTs were assessed using algometry in the thumb (dorsal aspect of the distal phalanx) and the midpoint of the trapezius muscle between the spinal processes of C7 and acromioclavicular joint three times every 30 s. The average was used for analysis. 

#### 2.3.2. Conditioned Pain Modulation (CPM)

To explain the CPM paradigm, it was necessary to describe firstly another pain measure collected in both clinical trials but which was not used in this secondary analysis. Temporal summation (TS) could use a test to evaluate the presence of central sensitization phenomena in pain patients [27]. It was assessed by applying the algometer 10 times (pulses) at the previously determined PPT with a pressure increase rate of approximately 2 kg/s. It was assessed on the dorsal surface of the middle finger of the right hand, midway between the first and second distal joints, as well as at the middle of the upper trapezius muscle on the right-hand side. Participants rated the intensity and unpleasantness of the pain from the first, fifth, and tenth pulses on a numeric pain rating scale (NPRS; 0 = no pain to 10 = worst possible pain). This method of TS was based on the previous protocol and quantifies the neuronal excitability of the pain facilitation paths [28].

The conditioning stimulus was an occlusion cuff applied to the left arm inflated until “the first sensation of pain”, at a rate of 20 mmHg/s, and remained at this point for 30 s. The cuff inflation was modified until the participants felt an intensity of 3 out of 10 on a verbal pain scale. CPM was measured by replicants of the TS assessment explained before with the inflated cuff. CPM measured the descending pain inhibitory system and its efficacy was measured following previous studies [28].

Both studies replicated the same procedure for QST described above. Firstly, PPT was measured and, after 2 min, CPM was assessed following the Cathcart protocol [29]. For more details about the setting of each study, the original studies can be considered [23,24].

#### 2.3.3. Fibromyalgia Impact Questionnaire (FIQ)

The disability and physical impact of FM were measured using the Spanish version of the FIQ. The total FIQ scores rate from 0 to 100, where higher scores mean lower quality of life. The FIQ has demonstrated good psychometric properties and an internal consistency of 0.93 in the Spanish FM population [30]. An improvement of at least 30% has been identified as the minimum for a positive response to treatment [31]. 

Cutoffs of fibromyalgia severity were selected as in previous studies, using a FIQ total score <39 to represent a mild effect, scores ≥39 to <59 to represent a moderate effect, and a score ≥59 to represent a severe effect [32].

#### 2.3.4. Jenkins Sleep Scale (JSS)

The JSS is a short questionnaire that has shown reliable psychometric properties in FM and in alleviating pain symptoms. It includes 4 questions that assess how often and how severe sleep problems are, with responses ranging from 1 to 5, where higher scores signify more severe sleep issues [33].

#### 2.3.5. Pain Catastrophizing Scale (PCS)

The Spanish-validated version of the Pain Catastrophizing Scale was used to measure catastrophizing pain [34]. It contains 3 subscales and a total score ranging from 0 to 52, with greater total scores indicating more catastrophizing and more intensity of negative feelings and thoughts regardless of pain. The Spanish adaptation of the PCS has shown strong psychometric properties in fibromyalgia patients, achieving a Cronbach’s alpha of 0.87 [35]. 

#### 2.3.6. State-Trait Anxiety Inventory (STAI)

The Spanish version of the STAI was used to assess trait and state anxiety; there are 20 items for evaluating trait anxiety and 20 items for assessing state anxiety, with each item rated on a 4-point scale. The total scores rate from 0 to 80 (each subscale), where higher scores indicate higher levels of anxiety [36,37].

### 2.4. Statistical Analysis

The statistical analysis was conducted using SPSS version 28.0 software (SPSS Inc., Chicago, IL, USA). All patients with no missing data on the target variables of the present study were included. Demographic data and clinical characteristics of the patients were reported using descriptive statistics. To analyze possible differences in sensory testing between patients with high, moderate, or low fibromyalgia impact, a Kruskal—Wallis test was used because the distribution of the residuals did not follow a normal distribution. For post hoc multiple comparisons after non-parametric analysis of variance, Dunn’s test using Bonferroni’s adjustment was used. A *p*-value of 0.05 was considered significant.

To assess the associations between the different variables, a quantile regression analysis was used due to the non-normal distribution of the residuals and the presence of outliers. Age was included as a covariate because it can act as a confounder. The *p*-value necessary to consider a significant association was determined using the Bonferroni adjustment.

## 3. Results

### 3.1. Participants Characteristics

One hundred and twenty-nine women with fibromyalgia were included. Table 1 shows the characteristics of the patients. 

### 3.2. Association of Clinical Manifestations with Mechanical Sensory Tests

Patients with severe fibromyalgia impact, compared to mild and moderate, had significantly lower CPM efficacy assessed on the finger. Patients with moderate fibromyalgia impingement had significantly higher PPT at the trapezius than those with mild or severe fibromyalgia impingement. No differences were found in the PPT at the finger and CPM at the trapezius (Table 2).

Quantile regression analysis after Bonferroni adjustment only showed an association between finger CPM and the impact of fibromyalgia. Sleep quality was not associated with any psychophysical variable. The results of the association analyses between these variables are shown in Table 3.

### 3.3. Association of Clinical Manifestations with Psychological Factors

Analyses showed a significant association of state anxiety, trait anxiety, and pain catastrophizing with the impact of fibromyalgia. Only pain catastrophizing was significantly associated with sleep quality. See the results in Table 4.

## 4. Discussion

The current study investigated the association between QST (pressure pain threshold and conditioned pain modulation) and the psychological status of the patients with the functional state of the patient, disability, catastrophizing, psychological state, and sleep quality. This study had the aim to link the gap between research and clinical practice for patients with fibromyalgia, attempting to link the functional status of the patient with evidence on central mechanism pain tests. The CPM and PPT used in this article could be used in clinical practice as an additional tool to provide a more comprehensive evaluation for this type of patient, using just an algometer and an inflation cuff. 

Our results suggest that PPT and CPM are not statistically significantly associated with the clinical status of the fibromyalgia patients. The only exception found in this study was an association between the CPM in the finger and the fibromyalgia impact; nevertheless, is not strong enough to be clinically significant. Coppieters et al. showed that altered CPM was linked with impaired cognition measures, and indices of central sensitization have been also shown to be associated with other measurements of quality of life [38]. Other studies did not show a statistically significant difference between healthy controls and fibromyalgia patients in either CPM or PPT [10]. Therefore, it is possible that PPT and CPM may not be appropriate assessment methods for this type of patient in clinical practice. Furthermore, our results are in line with Plisinga et al. [14] and Nahman-Averbuch et al. [39]; neither of them finds an association between the psychological factors and the CPM paradigm. Based on our data, the phenomenon of CPM occurs at a remote point, giving importance to the central sensitization phenomena. These findings suggest the significant variability that exists between points when evaluating the endogenous pain pathways.

One important point is the association between the impact of the fibromyalgia and the quality of sleep, the catastrophizing, and both the state and risk of anxiety. Similar findings were observed by Vela and collabs, where fibromyalgia patients had poorer quality of sleep and life [10]. Furthermore, psychopathologies such as anxiety or depression have been reported in around 29% of the fibromyalgic population, concluding that women with fibromyalgia are at a higher risk of developing these mental conditions [40]. This population has been shown to be more likely to develop major mental issues [41]. Moreover, another study has shown the relationship between sleep and catastrophizing, where “previous night non-restorative sleep was associated with higher morning pain catastrophizing” for fibromyalgia patients [42]. The same relationship has been described by Lazaridou et al., where it is described that through an exercise program, these variables act similarly, improving in the same way after engaging in physical activity [43]. Therefore, we can affirm the importance of psychological variables in this type of patient and the significant role they play in the impact of fibromyalgia.

On the other hand, sleep quality did not show any relationship between the QST and either the PPT or CPM. Sleep quality has been shown as a predictor of pain in fibromyalgia patients and has been reported as a problem in 99% of the patients in a trial [44], and has previously been associated with the functioning of pain mechanisms [45]. Recently, a systematic review and meta-analyses have concluded that sleep problems are directly associated with the risk of developing chronic widespread pain; however, it cannot be stated that this relationship is bidirectional [46]. It seems clear in the scientific literature that, in FM patients, there is an alteration of sensory tests and sleep quality, but we cannot associate both identities.

The scientific literature presents considerable debate over the appropriate CPM protocol and the type of conditioning stimulus to be used [47]. It is necessary to continue researching a reliable method to provide clinicians with a useful tool that can be used as a biomarker of pain. Future research using other types of stimuli instead of the pressure cuff, such as thermal (cold or heat) or electrical stimulation, would be interesting. Other methods could be interesting since they have shown significant negative correlations above the pressure cuff [47]. 

### 4.1. The Clinical Implications

This study highlights the importance of the clinical status of the patient beyond the pain measurements. The use of the FIQ, PCS, STAI, and the JSS could be powerful tools for assessing and monitoring changes occurring in the treatment of fibromyalgia. Using these four questionnaires, which evaluate disability, catastrophizing, psychological state, and sleep, we can draw a comprehensive view of the patient’s condition and have an accurate evaluation of their quality of life. 

Having an initial assessment of these four aspects of the patient’s functional status can provide added value in clinical practice. Evaluating the critical points of patients allows us to tailor the treatment and refer them to other healthcare professionals if it is necessary. Moreover, these assessments are much more cost-effective than mechanical sensory tests, as they do not require any additional equipment and can be completed by the patient at home. 

However, it is important to have a tool that can be used in the clinic as a biomarker and for the evaluation of endogenous pain pathways. This tool must be affordable for all clinicians and needs to be standardized.

### 4.2. Limitations

This study presented several limitations. The sample size was not enough to extrapolate our conclusions to the fibromyalgia population. Our analysis did not include any healthy control group with which to compare the data analyzed. Combining two databases may introduce biases, despite using the same measurement method, and only patients who had complete data were used in this study (*n* = 129). This secondary analysis is a cross-sectional design and therefore, causality of the existing relationships cannot be assumed. 

Further studies are needed that can translate the tools used in pain research into clinical practice. We must offer updated and effective assessment and monitoring options so that healthcare professionals, who have a direct relationship with fibromyalgia patients, can develop the best treatment plan. Also, there is a gap in the scientific literature between the pain modulation and the functional status of the patient; therefore, more studies are needed in this line to resolve this problem. 

## 5. Conclusions

This study evaluated whether the clinical status of FM patients and QST could be effective tools for clinical assessment. Our findings indicate that QST is not significantly associated with the clinical status of FM patients. Instead, the impact of fibromyalgia is strongly linked to pain catastrophizing, sleep quality, and anxiety state, but not to mechanical sensory tests.

Comprehensive assessments using tools like the FIQ, PCS, STAI, and JSS provide a more accurate evaluation of patients’ conditions. These assessments are cost-effective and self-administered, and support a holistic approach to FM management.

Our study highlights that mechanical sensory tests like PPT and CPM may not be reliable biomarkers for clinical status in FM. However, psychological factors such as pain catastrophizing and anxiety are significantly associated with FM’s impact and sleep disturbances, underscoring their importance in management.

While QST may not be reliable for assessing FM clinical status, comprehensive psychological and cognitive assessments are essential. A multidisciplinary approach tailored to individual patient needs is crucial for effective FM management and improving quality of life.

## Figures and Tables

**Table 1 jcm-13-04834-t001:** Participants characteristics.

Fibromyalgia Patients (*n* = 129)					
Variable	Mean	SD	Median	P25	P75
Age (yrs)	53.34	8.58	53	48	58
FIQ	57.83	17.19	61.1	45.6	70.3
JSS	15.11	4.19	16	12	19
PPT finger	3.39	1.56	3.06	2.5	4
PPT trapezius	1.81	0.86	1.7	1.3	2.26
CPM finger	−0.47	1.84	0	−1	1
CPM trapezius	−0.51	2.06	−0.9	−1	1
PCS	22.81	10.14	23	15	31
SAI	31.14	12.75	30	22	42
TAI	40.47	12.53	41	32	50

SD = Standard Deviation; y = years; FIQ = Fibromyalgia Impact Questionnaire; JSS = Jenkins Sleep Scale; PPT = Pressure Pain Thresholds; CPM = Conditioned Pain Modulation; PCS = Pain Catastrophizing Scale; SAI = State Anxiety Inventory; TAI = Trait Anxiety Inventory.

**Table 2 jcm-13-04834-t002:** Differences in psychophysical variables as a function of fibromyalgia severity.

Variables	Mild (*n* = 17)	Moderate (*n* = 40)	Severe (*n* =72)	Differences between Groups		Post Hoc Comparisons
	Median (IR)	Median (IR)	Median (IR)	Test	*p*-Value	*p*-Value *(Bonferroni*)
PPT finger	3 (2.8, 3.36)	3.27 (2.85, 4.5)	3 (2.43, 3.85)	X^2^ = 4.35	0.113	Sev-mild: 1.000 Mod-mild: 0.443 Sev-Mod: 0.056
PPT trapezius	1.76 (1.54, 2.26)	2.02 (1.63, 2.6)	1.5 (1.18, 1.97)	X^2^ = 14.71	<0.001 *	Sev-mild: 0.113 Mod-mild: 0.557 Sev-Mod: <0.001 *
CPM finger	0 (0, 2)	0 (−1, 1)	−1 (−2, 0)	X^2^ = 14.55	<0.001 *	Sev-mild: 0.002 * Mod-mild: 0.348 Sev-Mod: 0.009 *
CPM trapezius	0 (−1, 1.5)	−1 (−2, 1)	−0.5 (−1, 0.15)	X^2^ = 0.67	0.715	Sev-mild: 0.638 Mod-mild: 0.703 Sev-Mod: 1.000

* Significant differences (*p* < 0.05); IR = Interquartile Range; PPT = Pressure Pain Thresholds; CPM = Conditioned Pain Modulation.

**Table 3 jcm-13-04834-t003:** Association of clinical manifestations with mechanical sensory tests.

Clinical Manifestations		Fibromyalgia Impact		Sleep Quality	
		Coefficient (95% CI)	*p*-Value (Unadjusted)	Coefficient (95% CI)	*p*-Value (Unadjusted)
**Mechanical sensory testing**	CPM finger	−2.87 (−4.28, −1.46)	<0.001 *	−0.27 (−0.80, 0.25)	0.304
	Age	−0.73 (−1.03, −0.42)	<0.001 *	−0.10 (−0.21, 0.02)	0.089
	CPM trapezius	−1.84 (−3.47, −0.20)	0.028	−0.03 (0.11, 0.05)	0.428
	Age	−1.00 (−1.39, −0.60)	<0.001 *	−0.06 (−0.10, −0.03)	0.001 *
	PPT finger	−2.34 (−4.43, −0.24)	0.029	−0.44 (−1.08, 0.20)	0.174
	Age	−0.73 (−1.11, −0.34)	<0.001 *	−0.05 (−0.16, 0.07)	0.393
	PPT trapezius	−4.30 (−8.07, −0.54)	0.025	−0.01 (−0.05, 0.02)	0.432
	Age	−0.79 (−1.16, −0.40)	<0.001 *	0.01 (−0.01, 0.03)	0.181

* Significant differences after Bonferroni adjustment (*p* < 0.006). CI = Confidence intervals; PPT = Pressure Pain Thresholds; CPM = Conditioned Pain Modulation.

**Table 4 jcm-13-04834-t004:** Association of clinical manifestations with psychological factors.

Clinical Manifestations		Fibromyalgia Impact		Sleep Quality	
		Coefficient (95% CI)	*p*-Value (Unadjusted)	Coefficient (95% CI)	*p*-Value (Unadjusted)
**Psychological factors**	STAI-S	0.39 (0.14, 0.63)	0.002 *	0.08 (0.01, 0.16)	0.047
	Age	−0.71 (−1.08, 0.33)	<0.001 *	−0.06 (−0.19, 0.05)	0.254
	STAI-R	0.42 (0.16, 0.68)	0.002 *	0.02 (−0.06, 0.10)	0.581
	Age	−0.80 (−1.18, −0.41)	<0.001 *	−0.06 (−0.17, 0.05)	0.276
	PCS	0.54 (0.22, 0.86)	0.001 *	0.16 (0.08, 0.24)	<0.001 *
	Age	−0.58 (−0.96, −0.20)	0.003 *	−0.01 (−0.10, 0.08)	0.821

* Significant differences after Bonferroni adjustment (*p* < 0.008). CI = Confidence intervals; PCS = Pain Catastrophizing Scale; STAI = State-Trait Anxiety Inventory.

## Data Availability

Data are contained within the article.

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
