# Peer review of "The Association between Pressure Pain Thresholds, Conditioned Pain Modulation, Clinical Status, and Sleep Quality in Fibromyalgia Patients: A Clinical Trial Secondary Analysis"

_jcm, 2024, doi:10.3390/jcm13164834_

Round 1

Reviewer 1 Report

Comments and Suggestions for Authors

Dear colleagues, congratulations for ayour work and efforts in improving the management of FM patients.

Few minor suggestions:

1. row 81-82-83 - there is a repetition of the same phrase  (”leaving a gap in understanding regarding the potential modulatory effects of FM specific symptomatology”)

2. row 237 - the number of the Table is 3

3. row 245 - the number of the Table is 4

4. row 249 - please explain QST (Quantitative Sensory Test)

5. row 276 - anxiety and depression are not mental illnesses, in my opinion...More appropriate would be ”mental conditions”

Comments on the Quality of English Language

Minor revisions 

Author Response

We appreciate the opportunity to revise our manuscript jcm-3136809, updated and now entitled “The Association between Pressure Pain Thresholds, Conditioned Pain Modulation, Clinical Status, and Sleep Quality in Fibromyalgia Patients: A Clinical Trial Secondary Analysis” to JCM. We trust that you will find the current version informative to your readership and acceptable for publication. 

Thanks for your review commentaries to improve the quality of the manuscript. A deep and substantial modification has been carried out according to your suggestions.

Modifications to the manuscript text are denoted by line numbers and are written in blue in our manuscript. Please find our responses to each reviewer’s comment below. 

Sincerely,        

The authors. 

Comments 1: [row 81-82-83 - there is a repetition of the same phrase (”leaving a gap in understanding regarding the potential modulatory effects of FM specific symptomatology”)]

Response: Thank you very much for highlighting this mistake. We agree with this comment and we deleted one of the sentences.

Comments 2. row 237 - the number of the Table is 3

Response: Thank you for pointing this out. We have, accordingly, modified the number.

Comments 3. row 245 - the number of the Table is 4

Response: Thank you for pointing this out. We have, accordingly, modified the number.

Comments 4. row 249 - please explain QST (Quantitative Sensory Test)

Response: Thank you for this comment. We specified which QST we studied in order to clarify the discussion and also, we added a brief definition of QST in the introduction.

Comments 5. row 276 - anxiety and depression are not mental illnesses, in my opinion...More appropriate would be ”mental conditions”

Response: Thank you very much for your comment, “illnesses” has been changed to “mental conditions”

Reviewer 2 Report

Comments and Suggestions for Authors

Some suggestions for changes and improvements to be included in the document are as follows: 

- It is necessary to include information about the QST in both the abstract and the introduction with their respective bibliographic references.

- explain on the basis of which criteria the measurements on the finger and trapezium were made in the CPM

- it is necessary to clearly specify the different degrees of severity in fibromyalgia with their corresponding bibliographic citation.

- I consider that it is necessary to add some more valuable data to this study, since there is some information that has already been contrasted in other studies, so it is suggested that other data of interest be added that can surely be associated with the study of this study population. 

Author Response

We appreciate the opportunity to revise our manuscript jcm-3136809, updated and now entitled “The Association between Pressure Pain Thresholds, Conditioned Pain Modulation, Clinical Status, and Sleep Quality in Fibromyalgia Patients: A Clinical Trial Secondary Analysis” to JCM. We trust that you will find the current version informative to your readership and acceptable for publication. 

Thanks for your review commentaries to improve the quality of the manuscript. A deep and substantial modification has been carried out according to your suggestions.

Modifications to the manuscript text are denoted by line numbers and are written in blue in our manuscript. Please find our responses to each reviewer’s comment below. 

Sincerely,        

Tha authors.

- It is necessary to include information about the QST in both the abstract and the introduction with their respective bibliographic references.

Response: Thank you for this comment. We specified which QST we studied in order to clarify the study and also, we added a definition of QST in the introduction.

- explain on the basis of which criteria the measurements on the finger and trapezium were made in the CPM

Response: Thank you for your appreciation. We clarified that the measurements was at the baseline in the section “2.1 Study Design” and also, we added extra information in the “2.3.2 Conditioned pain modulation (CPM)”. 

- it is necessary to clearly specify the different degrees of severity in fibromyalgia with their corresponding bibliographic citation.

Response: Thank you very much for highlighting this mistake. We agree with this comment and we added more information about the cutoff and its bibliography in the “2.3.3 Fibromyalgia Impact Questionnaire (FIQ)” section.

- I consider that it is necessary to add some more valuable data to this study, since there is some information that has already been contrasted in other studies, so it is suggested that other data of interest be added that can surely be associated with the study of this study population.

Response: Thank you very much for your feedback. We agree that there is already literature on the subject, and we wanted to corroborate some of those hypotheses. However, this study aimed to bring pain research methods used under laboratory conditions closer to clinical practice, attempting to integrate these assessments into routine clinical practice.

Reviewer 3 Report

Comments and Suggestions for Authors

Thank you for inviting me to review the manuscript “The association between pressure pain thresholds, conditioned pain modulation, clinical status, and sleep quality in fibromyalgia patients: a clinical trial secondary analysis”.

This is a secondary analysis of a previous clinical trial on 129 patients with FM. 

In the abstract, the authors group the assessment tools of  FIQ, PCS, STAI as “clinical status”. I believe that, while the FIQ can be categorized as “clinical status”, the PCS and the STAI assess psychological outcomes. I think the authors should include also “psychological” outcome/functioning to more comprehensively describe the variables of interest (line 33-34).

The rest of the abstract is well written and summarized. 

Introduction:

Line 64-66: reference is needed 

Line 84: here the authors use the terminology of “clinical status”. I don’t argue with this choice, but I am wondering if the authors are referring here to the disease severity? I am ok in leaving “clinical status”; it is just that I don’t have a good grasp of what the authors mean with this term. 

Line 84-90: it is my understanding that the authors are providing a comprehensive list of factors influencing the outcome of the CPM tests. Is this correct? If that is the case, I suggest reiterating through the following rows that the outcome are related to CMP (CMP outcomes may be enough); otherwise, the reader is prone to believe that we are talking of the FM outcomes, not remembering that we are addressing CPM outcomes. 

Methods: more details are necessary to understand the design of the study. I understand that this is a secondary analysis, but it is not clear what the setting was. I guess this was originally a cross-sectional study where patients with FM have been investigated on those variables? But more (coincise) details are needed; e.g., who performed the experiments; always the same experimenter; if this was done at baseline or in the course of treatment; etc. I suggest doing this; or at least provide reference to the original study with a citation. 

Statistical analysis: 

Here the authors refer to FM patients classified into “high, moderate, or low FM impact”. However, this classification needs to be clarified either in the 2.3.3 FIQ subsection, or here in the statistical analysis (in other words, which cut-off are utilized to classify in these  three categories?)

Results: “Table 2” is repeated 3 times.

Discussion: well summarized

Limitations: line 320-322: repeated terms and the sentence lacks the verbe 

The original sample size of the entire database should be clarified (otherwise it was my understanding that the N = 129 corresponded to the entire database)

Author Response

We appreciate the opportunity to revise our manuscript jcm-3136809, updated and now entitled “The Association between Pressure Pain Thresholds, Conditioned Pain Modulation, Clinical Status, and Sleep Quality in Fibromyalgia Patients: A Clinical Trial Secondary Analysis” to JCM. We trust that you will find the current version informative to your readership and acceptable for publication. 

Thanks for your review commentaries to improve the quality of the manuscript. A deep and substantial modification has been carried out according to your suggestions.

Modifications to the manuscript text are denoted by line numbers and are written in blue in our manuscript. Please find our responses to each reviewer’s comment below. 

Sincerely,        

The authors

Comment: In the abstract, the authors group the assessment tools of  FIQ, PCS, STAI as “clinical status”. I believe that, while the FIQ can be categorized as “clinical status”, the PCS and the STAI assess psychological outcomes. I think the authors should include also “psychological” outcome/functioning to more comprehensively describe the variables of interest (line 33-34).

The rest of the abstract is well written and summarized. 

Response: Thank you for giving us constructive feedback. We have added “psychological status”

Comment: Introduction:

Line 64-66: reference is needed 

Line 84: here the authors use the terminology of “clinical status”. I don’t argue with this choice, but I am wondering if the authors are referring here to the disease severity? I am ok in leaving “clinical status”; it is just that I don’t have a good grasp of what the authors mean with this term. 

Response: Thank you for your appreciation. The term “clinical status” of the patient refers to the overall health condition, including symptoms and signs. It involves the medical assessment of their physical and mental state.

Comment: Line 84-90: it is my understanding that the authors are providing a comprehensive list of factors influencing the outcome of the CPM tests. Is this correct? If that is the case, I suggest reiterating through the following rows that the outcome are related to CMP (CMP outcomes may be enough); otherwise, the reader is prone to believe that we are talking of the FM outcomes, not remembering that we are addressing CPM outcomes. 

Response: Thank you very much for your feedback. We modified this paragraph to make it clearer than before.

Comment: Methods: more details are necessary to understand the design of the study. I understand that this is a secondary analysis, but it is not clear what the setting was. I guess this was originally a cross-sectional study where patients with FM have been investigated on those variables? But more (coincise) details are needed; e.g., who performed the experiments; always the same experimenter; if this was done at baseline or in the course of treatment; etc. I suggest doing this; or at least provide reference to the original study with a citation. 

Response: Thank you for your appreciation. We clarified that the measurements was at the baseline in the section “2.1 Study Design” and also, we added extra information in the “2.3.2 Conditioned pain modulation (CPM)”. 

Comment: Statistical analysis: 

Here the authors refer to FM patients classified into “high, moderate, or low FM impact”. However, this classification needs to be clarified either in the 2.3.3 FIQ subsection, or here in the statistical analysis (in other words, which cut-off are utilized to classify in these  three categories?)

Response: Thank you very much for highlighting this mistake. We agree with this comment and we added more information about the cutoff and its bibliography in the “2.3.3 Fibromyalgia Impact Questionnaire (FIQ)” section.

Comment: Results: “Table 2” is repeated 3 times.

Response: Thank you for pointing this out. We have, accordingly, modified the number

Comment: Discussion: well summarized

Response: Thank you very much for your comment, we really appreciate it.

Comment: Limitations: line 320-322: repeated terms and the sentence lacks the verbe 

Response: Thank you very much for highlight this mistake

Comment: The original sample size of the entire database should be clarified (otherwise it was my understanding that the N = 129 corresponded to the entire database)

Response: Thank you very much for your feedback. We agree that there is a lack of information and we added more info in the “2.2 Participants” section about the total number of patients.

Round 2

Reviewer 2 Report

Comments and Suggestions for Authors

Thank you for answering the questions posed above. 

I would appreciate it if in future studies you could add more content to your reports, relating or providing new avenues of research in this disease so in need of investigation.